# Evaluation and Management of Urological Complications Following Pediatric Kidney Transplantation: Experience from a Single Tertiary Center

**DOI:** 10.3390/medicina60111754

**Published:** 2024-10-25

**Authors:** Maria Sangermano, Enrico Montagnani, Serena Vigezzi, Marco Moi, Alessandro Morlacco, Nicola Bertazza Partigiani, Elisa Benetti

**Affiliations:** 1Pediatric Nephrology Unit, Department of Women’s and Children’s Health, Padua University Hospital, 35127 Padua, Italy; maria.sangermano@aopd.veneto.it (M.S.); serena.vigezzi@aopd.veneto.it (S.V.); marco.moi@aopd.veneto.it (M.M.);; 2Urology Clinic, Department of Surgery, Oncology and Gastroenterology, University of Padua, 35124 Padua, Italy; alessandro.morlacco@unipd.it

**Keywords:** kidney transplantation, pediatric, urological complication, ureteral stricture, vesicoureteral reflux, endoscopic surgery

## Abstract

*Background/Objectives*: Kidney transplantation is the treatment of choice for children with end-stage renal disease (ESRD), but its outcome can be affected by urological complications, with incidence rates of 2.5–25%. The aim of this study was to evaluate the occurrence of urological complications and their management in a cohort of pediatric kidney transplant recipients. *Materials and Methods*: A retrospective analysis on 178 patients who received a renal transplant at our Pediatric Kidney Transplant Center between 2011 and 2023 was conducted. Demographic and clinical data were analyzed. Urological complications were categorized as early, intermediate, or late based on their onset time. *Results*: Out of 178 patients, 28 (15.7%) experienced urological complications. Most patients (61%) had a pre-existing uropathy. Early complications (7–30 days) were all obstructive, namely, ureterovesical junction obstruction and perirenal collections. Intermediate complications (1–3 months) comprised ureteral stenosis, symptomatic vesicoureteral reflux (VUR), and obstructive lymphocele. Late complications (>3 months) included symptomatic VUR and ureteral stenosis, with one case leading to ureteral rupture. Early complications were often detected due to acute graft dysfunction, while late ones were mainly identified during routine clinical, laboratory, or ultrasound follow-up. Urological complications requiring surgical or endoscopic therapy were 13.4%. Most ureteral stenoses were treated with initial endoscopic stents, followed by definitive surgery. VUR was treated with endoscopic correction with a high success rate (75%), while open surgery was reserved for cases where initial treatments failed or complications recurred. No clear correlations were found between patient characteristics and risk of urological complication. Urological complications required multiple diagnostic procedures and therapeutic interventions (+2.5 admissions in mean and approximately +EUR 24,000) compared to an uncomplicated post-transplant course. However, they did not significantly impact transplant outcomes, with a graft survival rate comparable to that of the control group. *Conclusions:* Regular post-transplant follow-up is crucial, especially for patients with known risk factors, to allow for timely detection and treatment of urological complications, avoiding detrimental effects on graft function and improving transplantation outcomes.

## 1. Introduction

Kidney transplantation has emerged as the leading treatment for children with end-stage renal disease (ESRD) due to its superior outcomes, cost-effectiveness, and lower mortality rates compared to dialysis [1]. Over the years, advancements in immunosuppressive therapy and transplant surgical techniques have improved graft survival rates. However, long-term success can be hindered by several complications arising from episodes of rejection, severe infections, or relapsing underlying disease. Additionally, urological complications can have adverse effects on the function, survival, and general well-being of both the patient and the graft.

The most common urological complications reported in the literature include ureteral obstruction; vesicoureteral reflux (VUR), either asymptomatic or complicated by urinary tract infections (UTIs); anastomotic leakage and/or urinary extravasation; and lymphoceles [2]. Urological complications can be immediate (such as urine leaks or lymphoceles) or delayed (such as ureteral strictures, symptomatic VUR, urolithiasis, or recurrent UTIs), and may often require surgical intervention or multiple procedures. Therefore, promptly identifying and diagnosing urological complications is essential for optimizing graft function [3].

Data regarding urological complications in pediatric kidney transplant recipients are very limited compared to in the adult population. Therefore, it is hard to obtain definitive incidence rates for each complication and to define risk factors [4]. Several studies on adults identified male gender, delayed graft function, abnormal pre-transplant voiding cystourethrogram (VCUG), repeat transplantation, obesity, multiple donor arteries, and atrophic bladder as predisposing conditions for urological complications after transplantation [5,6]. Posterior urethral valves appeared to be a risk factor for ureteral obstruction, possibly due to pre-transplant bladder ischemia, thickness, and collagen remodeling [7]. In a retrospective study on pediatric kidney transplant recipients, a history of previous abdominal surgery emerged as risk factors for the development of intra-abdominal collections after transplantation [8]. The objective of this study was to evaluate the occurrence of urological complications and their management in a cohort of pediatric patients who underwent renal transplantation at our Center from 2011 to 2023. The secondary endpoint was to identify potential risk factors in order to develop a clinical–instrumental approach for early diagnosis and treatment.

## 2. Methods

### 2.1. Study Design

We conducted a retrospective analysis of 178 patients who underwent renal transplantation at our Pediatric Kidney Transplant Center between January 2011 and April 2023. Post-operative urological complications included in the evaluation comprised ureteral or ureterovesical obstruction (both intrinsic and extrinsic) and VUR complicated by UTI, lymphoceles, nephrolithiasis, new-onset bladder dysfunction, vesical lesions, or ureteral rupture.

Based on the time of onset, urological complications were classified as early (occurring between 7 days and 1 month post-transplantation), intermediate (occurring between 1 and 3 months post-transplantation), and late (occurring more than 3 months post-transplantation).

### 2.2. Collected Data

Demographic data included age and gender, cause of ESRD, donor source (deceased or living), history of previous kidney transplants, type and duration of previous renal replacement therapy, mean time of follow-up, previous abdominal surgery, previous bladder dysfunction or BK virus infection defined as symptomatic infection, and histological evidence of BK virus-associated nephropathy or blood BK–DNA copies stable at higher than 10,000 cp/mL for at least 6 months. Surgical details such as mean ureteral tube dwell time were recorded. Follow-up was considered concluded upon the resumption of renal replacement therapy in the case of graft failure, defined as end-stage kidney disease after transplantation.

According to our practice, a trans-anastomotic external stent is inserted up to the renal pelvis, and a urinary catheter is placed during transplantation intervention. In a functioning graft, the ureteral tube is removed on day 7 and the vesical catheter on day 8 after transplantation. Dwelling time is prolonged until functional recovery in case of delayed graft function or upon specific clinical indications (e.g., the vesical catheter is maintained if vesical recycling is indicated by the Pediatric Urologist).

### 2.3. Thorough Pre-Transplantation Urological Assessment

In our center, we actively assess modifiable risks both pre- and post-transplant to facilitate prevention and early management. Therefore, candidates for kidney transplantation routinely undergo a comprehensive pre-transplant urological assessment during hospitalization for inclusion on the transplant waiting list. This evaluation encompasses abdominal ultrasound for all patients and further evaluation for selected cases, such as voiding cystourethrography (VCUG) for those with congenital anomalies of the kidney and urinary tract (CAKUT). Urodynamic testing is also performed on patients with pre-existing bladder dysfunction.

The findings from these assessments are systematically reviewed before transplantation by a pediatric multidisciplinary team that includes a transplant-dedicated nephrologist, anesthesiologists, urologists, surgeons, and radiologists. This collaborative framework not only facilitates a thorough evaluation of each patient’s urological status but also enables the development of an individualized management plan tailored to their specific needs. [9]. This comprehensive approach reflects our commitment to minimizing complications and optimizing long-term outcomes in pediatric kidney transplant recipients.

### 2.4. Outcome Measures

The primary outcome was the identification occurrence of post-operative urological complications and possible associated risk factors. Secondary outcomes included:1.Diagnostic Yield: Successful diagnosis of the urological complication and effectiveness of the intervention (surgical or conservative);2.Complication Rates: Incidence of early, intermediate, and late complications;3.Hospital admissions and costs of urological complications.

### 2.5. Diagnostic Work-Up

Post-transplant suspected urological complications are initially assessed using conventional ultrasonography. According to our protocol, daily ultrasound (US) is performed during the first five days post-transplant. Further USs are performed after the removal of the ureteral stent and bladder catheter. Further examination is performed before discharge. Subsequent USs are performed during follow-up hospital admissions at 6 months, 12 months, and annually thereafter. Additionally, US is used to aid in the differential diagnosis of the various causes of renal function decline observed during biochemical assessments [10,11].

Depending on the clinical and US findings, further diagnostic investigations are employed as appropriate, including sequential renal scintigraphy with 99mTc-MAG3 in the case of suspected obstruction and VCUG for the evaluation of possible VUR. Further investigations, such as antegrade pyelography following percutaneous nephrostomy, retrograde ureteropyelography post-cystoscopy, abdominal computed tomography (CT), and magnetic resonance urography (MR urography) are performed according to the indications of the pediatric urologist. In cases of obstructive perirenal fluid collections, fluid analysis is performed as needed. Furthermore, patients who develop ureteral or ureterovesical stenosis are investigated for BK virus infection by BK virus–DNA testing in urine, blood, and renal and/or ureteral biopsy. Our protocol provides weekly BKV–DNA assessments in the first month post-transplant, bi-weekly in the second month, monthly from month 3 to month 6, every three months until the end of the first year, and biannually thereafter. When BKV–DNA is positive, viremia is checked every 2–3 weeks to monitor viral replication. If a kidney biopsy is performed, testing for BK virus is also conducted on the tissue sample. BKV reactivation, which can lead to polyomavirus-associated nephropathy (PVAN), is a significant risk for graft dysfunction and loss.

UTI diagnosis is defined based on fever and/or abdominal pain accompanied by the presence of more than 10^5^ colonies per mL of a single microorganism from a midstream clean-catch urine sample, or more than 10^3^ colonies per mL from a urine sample obtained by catheterization. In the case of clinically suspected pyelonephritis but a negative urine culture, acute DMSA scintigraphy is performed during the febrile episode. Suspected cases of bladder dysfunction are further investigated using video urodynamics testing. According to follow-up, every patient with urological complications is evaluated together with a pediatric urologist in a specific outpatient clinic, and the most challenging cases are then discussed in a fortnightly meeting in order to decide together which management might be the most appropriate.

### 2.6. Statistical Analysis

Group comparisons for continuous variables were performed using the independent *t*-test or the Mann–Whitney test, as appropriate (using the Kolmogorov–Smirnov test to assess the normality of the data distribution). Categorical variables were analyzed using the Chi-squared test and Fisher’s exact test. Statistical significance was defined as a *p*-value < 0.05. Continuous variables are presented as mean (standard deviation), while categorical variables are shown as counts and percentages. Correlation analyses, including both Pearson and Spearman coefficients, were conducted to explore the relationships between urological complications and the clinical variables. Logistic regression analysis was also performed to identify potential predictors of urological complications. The predictor variables entered into the regression model included sex, age, donor type, CAKUT, PUV, dialysis, duration of dialysis, bladder dysfunction, previous abdominal surgery, BK infection, and number of transplants. Data were analyzed using SPSS software 25.

We also compared the Kaplan–Meier survival time and graft survival time of the study group to the control group. In addition, a log-rank test statistic was used to highlight possible statistical differences between these two groups.

### 2.7. Cost Analysis

The cost of additional hospital admissions related to diagnostic and therapeutic urologic procedures, based on diagnosis-related groups (DRGs), was evaluated in our patients with urological complications compared to the control group.

## 3. Results

### 3.1. Characteristics of the Population

Among 178 pediatric recipients transplanted at our center between January 2011 and April 2023, 28 patients (15.7%) developed post-transplant urological complications. Children with urological complications did not differ from those without complications (150 patients, 84.3%) for demographic and clinical characteristics (gender, age, donor type, cause of ESRD, previous renal replacement therapy, number of transplants). The mean follow-up was 65 months (range 4–150 months).

Table 1 presents a detailed overview of the demographic characteristics of the group with post-transplant urological complications compared to the group without such complications.


medicina-60-01754-t002_Table 2Table 2Characteristics of the study group and comparison among ureteral obstruction and VUR subgroups.VariableStudy PatientsUreteral Obstruction Subgroup *VUR SubgroupNumber of patients2816 (57%)12 (43%)Gender


Male17 (60%)9 (56%)8 (67%)Female11 (40%)7 (44%)4 (33%)Cause of ESRD


CAKUT17 (61%)11 (69%)6 (50%)Renal hypoplasia/dysplasia/agenesis +/− VUR15103PUV312Urogenital sinus malformation101Other causes11 (39%)5 (31%)6 (50%)Congenital nephrotic syndrome422SRNS220Cystic disease413Renal impairment after neonatal asphyxia101Age at RTx, mean years (SD)11 (5)12 (5)8 (5)Source


Living donor8 (29%)4 (25%)4 (33%)Deceased donor20 (71%)12 (75%)8 (67%)Renal replacement therapy


Previous RTx and dialysis6 (21%)6 (37,5%)0 (%)Dialysis pre RTx[mean months, SD]13 (46%)[38.6, 27]6 (37,5%)[37.7, 25.7]7 (58%)[43.6, 30.6]Pre-emptive RTx9 (32%)4 (25%)5 (42%)Pre-existing bladder dysfunction8 (28%)5 (31%)3 (25%)Abdominal surgery pre-RTx15 (53%)8 (50%)7 (58%)Follow-up, mean months (SD)65 (36)60 (32)73 (41)Graft failure211VUR, vesicoureteral reflux. ESRD, end-stage renal disease. CAKUT, congenital anomalies of kidney and urinary tract. PUV, posterior urethral valves. SRNS, steroid resistant nephrotic syndrome. RTx, renal transplantation. SD, standard deviation. * Including extrinsic causes of stenosis (i.e., obstructive lymphocele and hematoma).


### 3.2. Classification and Management of Urological Complications

Among the children with urological complications (28 patients, 15.7%), 12 (6.7%) were symptomatic VUR, whereas 16 patients (9%) presented with ureteral or ureterovesical junction (UVJ) obstruction, also including obstruction due to extrinsic compression (2 cases of symptomatic lymphocele and 1 case of symptomatic hematoma). Detailed characteristics of patients with obstructive complications and VUR are summarized in Table 2.

Most children with urological complications (26 subjects, 92.8% of the study group) needed elective and/or emergency open, endoscopic, and/or minimally invasive surgery, yielding an incidence of complications needing surgery of 14.6% in the overall transplant population.

Drainage tube dwell time was available in 19 cases: Ureteral stent mean dwell time was 10 days (range 7–27 days) in patients with obstruction/urinary stenosis and 9.8 days in patients with VUR (range 7–13 days), without a statistically significant difference between the two groups (*p* = 0.93).

As regards the time of occurrence of complications, 7 cases (25%) were early, 9 cases (32%) were intermediate, and 12 cases (43%) were late complications. The type of surgical complications, time of occurrence, diagnostic studies conducted, and treatment performed are detailed in Table 3, Table 4 and Table 5.

The choice of treatment was determined on a case-by-case basis by a multidisciplinary team (including nephrologists, urologists, radiologists, and nuclear medicine specialists), considering the patient’s specific characteristics at the onset of the complication and the initial clinical presentation (e.g., acute kidney injury, reduced urine output). The selected treatment was aimed at minimizing the risk of complications while preserving graft function, in line with established literature guidelines and ensuring the best option for the patient. In our experience, the initial approach to ureteral stenosis involved ureteral stenting, which successfully resolved the issue in five cases (38.5%). In 61.5% of our cases, a definitive surgical correction, such as ureteral reimplantation or ureteroureterostomy, was required, with no recurrence of complication after the definitive treatment.

All 12 cases of VUR presented as intermediate or late complications occurring 23 months post-transplant. Symptoms were represented by UTI, with most patients (66.6%) presenting two or more UTIs. Table 6 shows all the bacteria identified in urine cultures: E. Coli and K. Pneumoniae, accounting respectively for 8 (44%) and 3 cases (17%) out of 18, were prevalent.

Most cases of VUR (8 out of 12, 4.5% of the overall population) were treated with endoscopic Deflux injections at the bladder level. Six patients had no recurrence of infection after the initial treatment, while two needed a second injection, with a 75% success rate of the first endoscopic treatment in preventing infection recurrence.

Diagnostic methods included contrast-enhanced voiding cystourethrography (ceVUS) (performed in one case at another hospital due to unavailability of standard voiding cystography) and DMSA scintigraphy. First-line treatment was endoscopic correction with Deflux injection in 8 out of 12 cases (67%). Two cases required repeated Deflux injections because of recurrent UTIs and VUR. Two children with low-grade reflux were managed conservatively with antibiotic prophylaxis. Circumcision was performed during the endoscopic procedure in two cases.

Bladder outlet obstruction and/or low bladder capacity were identified as the main causes of VUR in two children, who underwent percutaneous suprapubic cystostomy and bladder augmentation surgery with catheterizable stoma creation (Monti procedure).

### 3.3. Risk Factor Analysis for Urological Complications

The non-parametric Mann–Whitney U tests revealed no significant differences in the distributions of age and dialysis duration between patients with and without urological complications (*p* = 0.799 and *p* = 0.114, respectively). Similarly, the length of follow-up was similar between the two groups (*p* = 0.712). Correlation analysis showed a significant Pearson correlation between urological complications and living donor type (r = −0.179, *p* = 0.017), and a significant positive correlation with BK infection (r = 0.202, *p* = 0.007) and second kidney transplants (r = 0.241, *p* = 0.001).

The logistic regression model identified living donor type (B = −2.660, *p* = 0.024, OR = 0.070) and pre-transplantation dialysis (B = −1.881, *p* = 0.003, OR = 0.152) as possible protecting factors against urological post-transplant complications, while BK infection (B = 2.040, *p* = 0.005, OR = 7.689) was the only significant predictor of urological complications. Sex, age, dialysis before transplantation and its duration, underlying CAKUT or posterior urethral valve disease, previous bladder dysfunction, number of transplants, and previous abdominal surgery are not associated with urological complications, according to our model. The model’s overall classification accuracy was 84.3%, with a Cox and Snell R-squared of 0.147 and a Nagelkerke R-squared of 0.252, indicating a moderate explanatory power of the model (Table 7).

### 3.4. Costs of Urological Complications

Children who experienced urological complications required an average of 2.5 additional hospital admissions compared to those who did not. The median cost per admission at our center for the aforementioned diagnostic and therapeutic procedures was EUR 9574. Therefore, the average cost per patient increased by approximately EUR 23,935 in the presence of urological complications.

### 3.5. Patient and Graft Survival

We compared overall patient and graft survival between the group of children with urological complications and the control group. Survival curves are comparable between the two groups, with no statistically significant differences (Figure 1) in terms of overall survival (*p* = 0.40) and graft survival (*p* = 0.25).

## 4. Discussion

In our study, we found that 15.7% of pediatric kidney transplant recipients experienced urological complications, with obstructive issues in 9% and symptomatic vesicoureteral reflux (VUR) in 6.7% of the cases. This incidence is consistent with previously reported rates of 10–20% among pediatric kidney transplant recipients. In contrast, larger registry-based studies, such as those from the North American Pediatric Renal Trials and Collaborative Studies (NAPRTCS) database, report urological complications in approximately 8–13% of pediatric transplant patients. However, the literature reveals a broader range of overall incidence rates, from 2.5% to 25%. These discrepancies may arise from variations in surgical techniques, differences in follow-up durations, and inconsistent definitions of complications across studies, all of which can influence the observed rates [12,13,14,15,16,17,18,19].

More in detail, obstructive complications occurred in 9% (7.3% ab intrinseco and 1.7% ureteral compression by fluid collections) and symptomatic VUR in 6.7% of our population. In several recent pediatric series, ureteral stenosis ranged from 5% to 8% [14,15,16,17,18,19], and in a very large case series of 526 pediatric kidney transplants from a single tertiary center, ureteral obstruction occurred in 8% of cases, a rate that is consistent with our finding of 7.3% of patients experiencing post-transplant ureteral strictures ab intrinseco [20]. Therefore, ureteral stenosis emerges as the most common urological complication following kidney transplantation, predominantly occurring within the first year post-surgery, as confirmed by our series, in which all the ureteral or UVJ strictures were detected within 8 months from the transplant, with over 80% occurring within the first 3 months and a median time to diagnosis of approximately 2 months (58 days) post-transplantation.

In our study, 6.7% of the patients developed symptomatic vesicoureteral reflux (VUR), with 83.3% of these cases requiring endoscopic or surgical intervention. This incidence falls within the range reported in the literature, where post-transplant symptomatic VUR rates have been documented between 2% and 13%. The need for surgical or endoscopic treatment in the majority of our cases aligns with previous findings, which highlight that a significant proportion of VUR cases, especially those that are symptomatic, often require more invasive management to prevent recurrent urinary tract infections and preserve graft function [18].

Taking into consideration the risk factor analysis, the results of previous studies identified CAKUT, BK virus infection, posterior urethral valves, and a history of prior ureteral surgery as risk factors for urological complications in pediatric kidney transplant recipients together with donor and surgical factors [12,13,17,18,19].

We aimed to identify potential risk factors for such complications in our cohort by comparing patients who had developed urological complications with a control group of recipients. Our analysis highlighted the possible role as a risk factor of BK infection and the presence of a posterior urethral valve for developing urological complications. This is consistent with the other reports from the existing literature (Refs. [3,12,13,17,18,19,21]). A total of 28% of our patients had pre-existing bladder dysfunction, with 25% having both lower urinary tract dysfunction (LUTD) and CAKUT as the underlying cause of ESRD. LUTD has already been demonstrated to be a significant risk factor for developing post-transplant urological complications [12,14]. These data emphasize the importance of a comprehensive urological evaluation during pre-transplant assessment, especially in children with such risk factors.

Optimal evaluation and management strategies for these patients remain uncertain; however, recommendations include abdominal US, voiding cystourethrography for patients with a history of UTIs or prior urological interventions, and invasive urodynamic studies for bladder dysfunction [22,23,24]. Management of lower urinary tract issues may involve bladder rehabilitation therapy, pharmacological treatments, intermittent catheterization, and surgical reconstruction. Conservative treatments to improve native bladder function should be prioritized before reconstructive procedures such as urinary diversion or bladder augmentation. Reconstructive surgery is generally recommended before transplantation, but caution is advised in children with severe chronic kidney disease [25].

On the contrary, our logistic regression model identified living donor type and pre-transplantation dialysis as possible protective factors against urological post-transplant complications. This last finding contrasts with some previous reports, but the differences may be due to the specificity of our pediatric population and the minor quantity of time in dialysis [6]. We could not stratify for type of dialysis because of the small overall sample size. Therefore, this difference is difficult to explain in light of the available data. A more specific analysis for type of dialysis and time spent in dialysis might cast some light on the discrepancy from the literature.

The fact that living donor type is protective is less surprising because of the careful and thorough pre-transplant assessment before proceeding to living-donor transplantation. Previous studies have, however, not identified this protective role [14].

There were no statistical differences in terms of urological complications between our populations in terms of sex, age, length of dialysis before transplantation, underlying CAKUT, previous bladder dysfunction, number of transplants, and previous abdominal surgery. Previous research also reported no notable differences in obstruction rates based on age or gender. Some authors attributed this to the relative rarity of these complications, which may limit the ability to detect statistically significant differences [19,24].

In our population, urological complications requiring surgical or endoscopic therapy comprised 13.4%. In a recent retrospective analysis of 136 pediatric patients undergoing kidney transplantation, the incidence of urological complications requiring surgical or endoscopic therapy for stenosis or VUR was 13% [17]. Taher et al. reported that nearly 20% of 146 pediatric patients undergoing kidney transplantation experienced intra-abdominal complications, and about 6.2% required surgical drainage for fluid collections [8,24].

Endourological techniques, such as balloon dilation and stenting, are now the standard procedure for managing ureteral obstructions, thanks to their minimally invasive nature and high success rates. These methods are particularly effective for stenoses under 1.5 cm and for treatment within 3 months of onset [26]. In our study, the initial approach to ureteral stenosis involved ureteral stenting, which successfully resolved the issue in five cases (38.5%). Success rates reported in the literature are generally higher, but the difference may be attributed to the length of the stenosis and to the timing of intervention [27]. In 61.5% of our cases, a definitive surgical correction, such as ureteral reimplantation or ureteroureterostomy, was required. This is consistent with other studies, which highlight that more complex stenoses or failures of endoscopic treatment often necessitate surgical intervention [28]. Endoscopic treatment with submucosal Teflon injection (Sting) is widely used for the treatment of VUR because it is easily performed and is associated with low morbidity. Success rates are between 54 and 74% overall and can reach up to 90% in patients with low-grade VUR [29]. The procedure can also be repeated in patients who do not show improvement after a single session. In line with the surgical standard, most cases of VUR (8 out of 12, 4.5% of the overall population) were treated with endoscopic Deflux injections at the bladder level [30]. Six patients had no recurrence of infection after the initial treatment, while two needed a second injection, with a 75% success rate of the first endoscopic treatment in preventing infection recurrence.

The literature reports a 50% success rate with Deflux for VUR management and suggests that endoscopic treatment is more effective in patients without lower urinary tract dysfunction, while it shows higher failure rates in those with such issues [13,31].

Importantly, these complications required multiple diagnostic procedures and therapeutic interventions, with an overall increase in hospitalization episodes (+2.5 admissions in mean) and costing approximately EUR 24,000 compared to an uncomplicated post-transplant course. However, urological complications did not significantly impact transplant outcomes, with a graft survival rate comparable to that of the control group. This could be attributed to a careful screening for post-transplant urological complications, which allowed for timely detection and treatment, contributing to preserving graft function.

Additionally, the advancement of minimally invasive techniques facilitates more effective management of these complications, further contributing to the preservation of graft function. Previous studies have already demonstrated that urological complications do not significantly impact graft survival [32]. In a study by van Roijen et al. involving 695 kidney transplant recipients, 10.8% developed urological complications. However, the graft failure rates at 5 and 10 years were comparable between patients with and without these complications, suggesting that early identification and treatment of urological complications do not compromise graft function [32,33].

These findings highlight the importance of early recognition and management of urological complications in pediatric kidney transplant recipients. While these complications increase morbidity and healthcare resource use, they do not appear to compromise long-term graft outcomes if appropriately addressed. Continued advances in endourological and surgical techniques offer promising options for minimizing the impact of these complications, further improving the overall success of pediatric kidney transplantation.

The major strength of this study is that it relies on the numbers of an exclusively pediatric population taken in a high-volume transplant center such as ours. However, there are some weaknesses that need to be addressed. The retrospective observational design of our study and the single-center data may not make them standardizable and generalizable, owing to possible differences in terms of diagnostic and therapeutic work-up.

Taking into consideration some future perspectives in terms of ameliorating graft survival and the incidence of complications, we think that building a prospective and up-to-date complication registry will help to describe the overall incidence of complications and overcome biases due to the differences between centers.

## 5. Conclusions

Urological complications are common after pediatric kidney transplantation, with an overall incidence and risk factor profile consistent with the few reports already available in the literature. Pre-transplant comprehensive urological assessment can help in minimizing post-transplant complications and can guide patient-tailored surgical techniques. Meticulous post-transplantation follow-up and collaboration with a well-coordinated multidisciplinary team may allow for timely detection and treatment of complications, avoiding detrimental effects on graft function and improving transplantation outcomes.

## Figures and Tables

**Figure 1 medicina-60-01754-f001:**
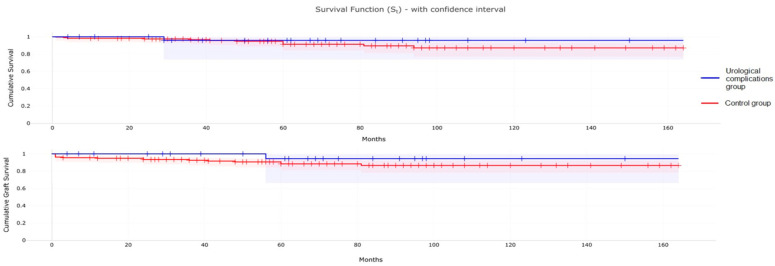
Overall patient (**up**) and graft survival (**down**) curves in recipients with urological complications (blue line) and in the control group (red line).

**Table 1 medicina-60-01754-t001:** Characteristics of the patients.

Variable	Study Group	Control Group
Number of patients	28	150
Gender		
Male	17 (60%)	99 (66%)
Female	11 (40%)	51 (34%)
Source		
Living donor	8 (29%)	26 (17.3%)
Deceased donor	20 (71%)	124 (82.7%)
Mean age at RTx, y (SD)	11 (5)	9.61 (5.57)
Cause of ESRD		
CAKUT	17 (61%)	73 (48.6%)
Other causes *	11 (39%)	77 (51.4%)
Renal replacement therapy		
Previous RTx and dialysis	6 (21%)	17 (11.3%)
Dialysis pre RTx	13 (46%)	106 (70.7%)
Pre-emptive RTx	9 (32%)	27 (18%)

RTx, renal transplantation. SD, standard deviation. ESRD, end-stage renal disease. CAKUT, congenital anomalies of kidney and urinary tract. * Detailed in Table 2.

**Table 3 medicina-60-01754-t003:** Early urological complications (occurring between 7 and 30 days after transplantation).

Complications	N	Post-RTx Onset Day	Diagnostic Studies	First Treatment	Subsequent Treatments
UVJ obstruction (AKI) + urinary leakage	3	8th (2), 11th	Ultrasound (US)(3), anterograde pyelography (2); cystoscopy + retrograde pyelography (1),	Emergency nephrostomy and DJ stent placement (2); MJ ureteral stent placement (1)	DJ replacement (1), ureteroneocystostomy (3)
Obstructive lymphocele	1	21st	US, drained fluid analysis	Surgical drainage into peritoneal cavity	
UVJ obstruction (AKI) + urinary leakage	2	21st, 26th	US (2), MAG3 scintigraphy (2), anterograde pyelography (1)	Ureteroneocystostomy and DJ stent placement (1), emergency nephrostomy (1)	DJ stent placement (1)
Obstructive perirenal hematoma	1	27th	US	DJ stent placement	
Total	7				

AKI, acute kidney injury. DJ stent, double-J stent. MJ stent, mono-J stent. RTx, renal transplantation. US, ultrasound. UVJ, ureterovesical junction.

**Table 4 medicina-60-01754-t004:** Intermediate urological complications (occurring between 30 and 90 days after renal transplantation).

Complications	N	Post-RTx Day of Onset	Diagnostic Studies	First Treatment	Subsequent Treatments
VUR complicated by UTI (3)	3	35th, 59th and 63rd	VCUG(3), DMSA scintigraphy (3)	Endoscopic injection of a bulking agent (Deflux) (2) + circumcision (1), conservative (antibiotic prophylaxis) (1)	2nd endoscopic injection of a bulking agent (Deflux) (2)
Obstructive lymphocele	1	42nd	Ultrasound (US), drained fluid analysis	US-guided percutaneous drainage	
Middle 1/3 ureteral stricture (AKI)	1	48th	US, MAG3 scintigraphy	DJ ureteral stent placement	End-to-side anastomosis of the transplant to the native ureter
Ureteral (1) and UVJ (AKI) (1) stricture in suspected BK virus infection/reactivation	2	65th and 79th	US (2), MAG3 scintigraphy (2), magnetic resonance urography + cystoscopy and retrograde pyelography (1)	DJ ureteral stent placement (2) + balloon dilatation (1)	Nephrostomy and subsequent end-to-side anastomosis of the transplant to the native ureter (1)
Urolithiasis and pyeloureteral junction stricture (AKI)	1	58th	US; anterograde pyelography; MAG3 scintigraphy; abdominal CT + transnephrostomic contrast study under CT guidance;	Emergency nephrostomy	DJ ureteral stent placement
Bladder outlet obstruction and UVJ stricture	1	69th	US, MAG3 scintigraphy	Vesicostomy	DJ stent placement and subsequent continent enterocystoplasty and catheterizable stoma (Monti procedure)
Total	9				

AKI, acute kidney injury. CT, computed tomography. DJ stent, double-J stent. RTx, renal transplantation. US, ultrasound. UTI, urinary tract infection. UVJ, ureterovesical junction. VCUG, voiding cystourethrogram. VUR, vesicoureteral reflux.

**Table 5 medicina-60-01754-t005:** Late urological complications (occurring more than 90 days after renal transplantation).

Complications	N	Post-RTx Day of Onset	Diagnostic Studies	First Treatment	Subsequent Treatments
Bladder dysfunction and VUR complicated by UTI	1	110th	US, MAG3 scintigraphy, video urodynamic tests	Suprapubic percutaneous cystostomy	Stimulation of the posterior tibial–pudendal nerve
VUR complicated by UTI in pre-existing poor bladder capacity	1	117th	US (1), ceVUS (1),	Bladder augmentation	
UVJ obstruction with ureteral rupture and abscessed urinoma. Recurrent UVJ obstruction after ureteral stent removal	1	157th	Abdominal CT, drained fluid analysis, retrograde endoscopic urethrography. anterograde pyelography and retrograde endoscopic urethrography.	US-guided percutaneous drainage, MJ ureteral stent placement. Emergency nephrostomy.	Replacement of ureteral MJ stent with DJ stent. Subsequent replacement of a transplanted ureter with a native ureter and ureteral stent placement.
Exophytic amorphous vesical lesion and UVJ stricture	1	199th	US, MAG3 scintigraphy, cystoscopy	Endoscopic removal of bladder exophytic lesion and placement of a DJ ureteral stent	2nd endoscopic removal of bladder exophytic lesion and placement of a DJ ureteral stent
Ureteral stricture	1	219th	Sonography, MAG3 scintigraphy	DJ ureteral stent placement	Uretero-neocystostomy, DJ stent placement and circumcision
VUR complicated by UTI (6), urethral stricture (1)	7	270th, 545th, 567th, 771st, 1168th, 1764th, 3121st	DMSA scintigraphy (5), VCUG (5), video urodynamic tests (1)	Endoscopic injection of a bulking agent (Deflux) (5), conservative (antibiotic prophylaxis) (1), endoscopic urethral dilatation (1)	Circumcision (1)
Total	12				

ceVUS, contrast-enhanced voiding urosonography. CT, computed tomography. DJ stent, double-J stent. MJ stent, mono-J stent. RTx, renal transplantation. US, ultrasound. UTI, urinary tract infection. VCUG, voiding cystourethrogram. VUR, vesicoureteral reflux.

**Table 6 medicina-60-01754-t006:** Bacteria identified as the cause of UTI affecting the transplanted kidney through urine culture in patients with VUR.

Patient Number	1st UTI	2nd UTI	3rd UTI
Patient No. 1	*Klebsiella Pneumoniae*	*Escherichia coli*	*Escherichia coli*
Patient No. 2	*K. Ornithinolytica*	*K. Ornithinolytica*	*Escherichia coli*
Patient No. 3	*Klebsiella Pneumoniae* and *Escherichia coli*	*Klebsiella Oxytoca*	
Patient No. 4	*Enterococcus faecium*	*P. Aeruginosa*	*P. Aeruginosa*
Patient No. 5	*Escherichia coli*		
Patient No. 6	*Enterococcus faecalis*		
Patient No. 7	*Escherichia coli*		
Patient No. 8	*Klebsiella Pneumoniae*	*Escherichia coli*	
Patient No. 9	*Escherichia coli*		

UTI, urinary tract infection.

**Table 7 medicina-60-01754-t007:** Logistic regression model results.

Variable	*p*-Value	Odds Ratio (OR)
Sex	0.845	0.908
Age	0.990	1.001
Living Donor	0.024	0.070
Congenital Abnormalities (CAKUT)	0.385	1.587
Posterior Urethral Valves (PUV)	0.091	0.223
Dialysis	0.003	0.152
Duration of Dialysis (months)	0.172	1.012
Bladder Dysfunction	0.438	1.665
Previous Abdominal Surgery	0.753	1.200
BK Virus Infection	0.005	7.689
Number of Trasplant	0.157	2.397

## Data Availability

Data are contained within the article; further inquiries can be directed to the corresponding author.

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
