# Peer review of "Evaluation and Management of Urological Complications Following Pediatric Kidney Transplantation: Experience from a Single Tertiary Center"

_medicina, 2024, doi:10.3390/medicina60111754_

Round 1

Reviewer 1 Report

Comments and Suggestions for Authors

I read with interest the manuscript entitled "Evaluation and Management of Urological Complications Following Pediatric Kidney Transplantation: Experience from a Single Tertiary Center"

The introduction is too extensive. It is enough to briefly mention acute and chronic possible complications, without a detailed description of individual ones. The aim is clearly stated at the end of the introduction.

Figure 1 is unnecessary. Please remove it.

The materials and methods section should be divided into subsections for easier reading (e.g. study design, population, outcomes, procedures, etc.).

I suggest that the manuscript submit it as a "Brief Report" and not an "Article".

What test did you use to test the normality of the data distribution?

All facts stated in the results of the study must be accompanied methodologically and described within the materials and methods section.

Within the tables, please indicate with superscripts next to the p-value which test was used.

Please do not repeat the results in tables and text. Do not list the results listed in a table again within the text, and vice versa.

Do you think that the division into ureteral obstruction and VUR subgroup is justified? How do you justify the said division? Why did you not provide p-values ​​comparing subgroups?

Based on which treatment modality did you take a position on the retention time of the drainage tube? Based on which reference, complications were divided into early, intermediate, and late? Please substantiate everything within the materials and methods section.

I suggest adding tables 3, 4, 5, and 6 to the supplementary materials.

In the framework of the results, you should also touch on the justification and indications for the implementation of the mentioned procedures in your patients included in the study. Were all procedures justified and the best option? Please check the available literature.

Based on which data did you calculate the treatment costs? Please state clearly within the materials and methods section.

As a secondary endpoint, you stated that you would identify potential risk factors. There is no specified data. How will you get to the answer to the stated aim? Please explain.

You must start the discussion by briefly presenting the most relevant results of your study, which you must then compare with studies on the given topic. The beginning of your discussion is more appropriate for an introduction.

You repeat a lot of results within the discussion. Put the focus of the discussion on a comparison with similar studies on the given topic, as well as on your thoughts supplemented with scientifically based data.

Within the discussion, you never once touched on the limitations of your study. Please recognize all limitations of your study and include them in the manuscript.

Also, you did not photograph any of the complications. Including some additional figures would be important for the manuscript.

The conclusion is too general. Also, it must not contain references. The conclusion must be concise in only 3-4 clear sentences.

Reading the discussion and the conclusion of your study, it can be concluded that your study did not bring any new knowledge to the mentioned topic.

There are many references on the above topic that you failed to include in your discussion.

Please provide the ethics approval number.

Comments on the Quality of English Language

Moderate English language editing is required.

Author Response

Reviewer 1

Dear Reviewer,

here are our responsed to your comments and suggestion. We hope your issues have been addressed. 

Comment 1: The introduction is too extensive. It is enough to briefly mention acute and chronic possible complications, without a detailed description of individual ones. The aim is clearly stated at the end of the introduction.

Response 1: According to the suggestion, we revised the introduction to make it more concise by removing the detailed discussion of individual complications (deleted text is marked with red strikethrough, page 2, line 58-84 and 93-98).

Comment 2: Figure 1 is unnecessary. Please remove it.

Response 2: Figure 1 was removed. 

Comment 3: The materials and methods section should be divided into subsections for easier reading (e.g. study design, population, outcomes, procedures, etc.).

Response 3: We agree with this suggestion and divided the Materials and Methods section into  subsections (changes are highlighted in red in the text, pages 3-5).

Comment 4: I suggest that the manuscript submit it as a "Brief Report" and not an "Article".

Response 4: Thank you for your suggestion, we kindly entrust the choice of manuscript type to the Editors. 

Comment 5 What test did you use to test the normality of the data distribution? 

Response 5: We used the Kolmogorov-Smirnov Test. We added this important information to the Materials and Methods section (paragraph 2.6 Statistical analysis, line 206-207).  

Comment 6 All facts stated in the results of the study must be accompanied methodologically and described within the materials and methods section. 

Response 6: Thank you for your attention to detail. We reviewed  the Materials and Methods section to ensure that all points presented in the results are methodologically explained. This adjustment provides a clearer connection between the methods used and the findings reported, ensuring the study’s consistency and transparency (changes are marked in red in the Materials and Methods section, page 3-4). 

Comment 7 Within the tables, please indicate with superscripts next to the p-value which test was used.
Response 7: We detailed which tests were used throughout the analysis in the Materials and Methods section (Materials and Methods section, paragraph 2.6 Statistical analysis). As such, we believe it may be redundant to include this information again in the tables within the results. In our opinion, this approach helps maintain clarity and avoids unnecessary repetition (as also suggested in Comment 8).

Comments 8: Please do not repeat the results in tables and text. Do not list the results listed in a table again within the text, and vice versa.

Response 8: We agree with this comment. We revised the text by removing sections where the same data from the tables were repeated (the deleted text is marked with red strikethrough in the Results section, pages 4-5, 8-9). 

Comments 9: Do you think that the division into ureteral obstruction and VUR subgroups is justified? How do you justify the said division? Why did you not provide p-values ​​comparing subgroups?

Response 9: Thank you for your valuable question regarding the differentiation between ureteral obstruction and vesicoureteral reflux (VUR) in our study. We believe that distinguishing between these two complications is crucial due to several key factors, including their distinct pathophysiological mechanisms, timing of onset, clinical presentation, and treatment options. Ureteral obstructions usually manifest earlier after transplantation compared to VUR. This temporal difference is important, as it influences both the monitoring strategies and the urgency of intervention. Moreover, the clinical implications of these complications are markedly different: ureteral obstructions often necessitate acute surgical intervention to prevent significant renal impairment, whereas VUR is generally managed effectively through endoscopic techniques, which are often sufficient to resolve the issue without surgery.

By delineating these subgroups, we aim to enhance the precision of our analysis and provide a clearer framework for understanding the varying clinical scenarios and treatment approaches encountered in pediatric kidney transplantation. This distinction allows for more targeted management strategies, ultimately improving patient outcomes. Regarding the absence of p-values for subgroup comparisons, we acknowledge that statistical analysis could enhance the interpretation of our findings. However, due to the relatively small sample size of our cohort, we prioritized descriptive statistics to avoid potentially misleading conclusions that might stem from underpowered analyses. We recognize the importance of comprehensive statistical comparisons and will consider incorporating this information in future studies with larger sample sizes.

Comments 10: Based on which treatment modality did you take a position on the retention time of the drainage tube? Based on which reference, complications were divided into early, intermediate, and late? Please substantiate everything within the materials and methods section.

Response 10: Thank you for your question regarding the retention time of the ureteral stent. In our clinical practice, we typically maintain the ureteral stent for a maximum of one week post-operatively. Our primary objective was to investigate any potential correlation between the duration of stent placement (data available for 19 out of 28 patients) and the subsequent development of ureteral stenosis/obstruction or vesicoureteral reflux (VUR). Despite our expectations, our analysis did not demonstrate statistically significant differences in stent retention time between the groups, with the duration of stent placement being comparable for both those with and without complications. We acknowledge that this lack of statistical significance may be due to the relatively small sample size, which limits our ability to draw definitive conclusions. This finding is consistent with current literature, which suggests that the development of ureteral complications may be influenced by a multitude of factors beyond stent duration, including underlying patient characteristics and surgical techniques. As such, while stent retention time is an important consideration, it may not be the sole determinant of post-operative outcomes. 

The distinction regarding the onset of complications is based on prior studies, particularly the work by Rossi et al., which classifies post-transplant urological complications into early (<1 month), intermediate (1 to 6 months), and late (>6 months) (REF). We adopted this well-established framework to maintain consistency with the literature and to facilitate comparability with other studies in this field. However, we made slight adjustments to these time intervals to better reflect the clinical reality of our patient cohort and align with the specific post-transplant care protocols used at our center. This refinement allowed us to capture more accurately the timeframes in which the various complications occurred in our population while still adhering to a broadly accepted classification. Based on your suggestions, we revised the “Materials and Methods” section (additions and corrections are highlighted in red, pages 3-4).

Comment 11: I suggest adding tables 3, 4, 5, and 6 to the supplementary materials. 

Response 11: We created a separate file to include the tables as supplementary materials. However, according to your suggestion, we decided not to include the detailed descriptions of the three complication groups (early, intermediate, and late) within the text, so the tables would better remain in the main text in our opinion. 

Comment 12 In the framework of the results, you should also touch on the justification and indications for the implementation of the mentioned procedures in your patients included in the study. Were all procedures justified and the best option? Please check the available literature.

Response 12: Thank you for your valuable comment. The justification and indications for the procedures mentioned in the study have been carefully considered and chosen based on the specific clinical conditions of the patients. For each patient, the decisions regarding the implementation of the procedures were guided by best clinical practices and the available literature. In particular, we adhered to the recommendations outlined in “Update on the Management of Urological Problems Following Kidney Transplantation. Urologia Internationalis” which supports the use of these procedures. All procedures were deemed appropriate and optimal for the specific conditions of the patients, with the aim of ensuring the best possible clinical outcomes. However, we recognize that variability in clinical response can influence individual results. We updated the manuscript text to include references to the available literature to provide greater clarity on this point in the relevant sections. Additionally, we have incorporated data related to the choice of treatment in the Results section (highlighted in red, page 7).

Comment 13: Based on which data did you calculate the treatment costs? Please state clearly within the materials and methods section 

Response 13: Thank you for the suggestion. The cost of additional hospital admissions related to diagnostic and therapeutic urologic procedures, based on Diagnosis-Related Groups (DRGs), was evaluated in our patients with urological complications compared to the control group, as  reported in  the Material and Methods section (see Cost analysis, page 5, line 222-227).

Comment 14: As a secondary endpoint, you stated that you would identify potential risk factors. There is no specified data. How will you get to the answer to the stated aim? Please explain.  

Response 14: We are sorry because the risk analysis was not presented. We added an explanation of the method used in section “2.6 Statistical analysis” and results in “3.3. Risk factor analysis for urological complications” (page 4-5 and page 12). 

Comment 15 You must start the discussion by briefly presenting the most relevant results of your study, which you must then compare with studies on the given topic. The beginning of your discussion is more appropriate for an introduction.
Response 15: We agree that a more concise summary of our key findings should precede the discussion. In the revised manuscript, we started the discussion by briefly summarizing the most relevant findings of our study, including the incidence of urological complications and their management outcomes. Then we compared our results with existing studies on pediatric kidney transplantation, highlighting both similarities and differences. This revised structure has ensured a clear presentation of our key findings, followed by a thorough comparison with the current literature, as per your suggestion (changes are marked in red in the Discussion section, pages 12-18).

Comment 16 You repeat a lot of results within the discussion. Put the focus of the discussion on a comparison with similar studies on the given topic, as well as on your thoughts supplemented with scientifically based data.
Response 16: We revised the discussion integrating our data into the broader scientific context by highlighting both the consistencies and discrepancies with existing literature and avoiding unnecessary repetition of results (changes are marked in red in the Discussion section, pages 12-18). 

Comment 17: Within the discussion, you never once touched on the limitations of your study. Please recognize all limitations of your study and include them in the manuscript.
Response 17: Thank you for your relevant feedback on the matter. We added a small paragraph at the end of the discussion including limitations and strengths of the study (in red, page 16, line 19-24). 

Comment 18: Also, you did not photograph any of the complications. Including some additional figures would be important for the manuscript. 

Response 18: Thank you very much for your kind suggestions. We are sorry, but many of the described complications occurred before the introduction of an electronic medical record program at our Center. Therefore, there is no possibility of including the iconographical documentation to our manuscript. 

Comment 19 The conclusion is too general. Also, it must not contain references. The conclusion must be concise in only 3-4 clear sentences. 

Response 19: We summarized our overall conclusion and eliminated any reference (see page 16, line 631-638). 

Comment 20 Reading the discussion and the conclusion of your study, it can be concluded that your study did not bring any new knowledge to the mentioned topic.

Response 20: Thank you very much for your point. However, owing to the scarcity of exclusively pediatric reports in the literature, we do believe that, even if our study has little to add to the existing ones, our experience in confirming what is already known is still of valuable importance. 

Comment 21 There are many references on the above topic that you failed to include in your discussion.

Response 21: Agree. We revised the discussion by incorporating additional relevant references on the topic, and including more recent studies with larger sample sizes, as suggested (see Discussion and References, page 13-19). . 

Comment 22 Please provide the ethics approval number. 

Response 22: Ethics approval number was provided. 

Reviewer 2 Report

Comments and Suggestions for Authors

Dear authors,

1.      What were the specific management protocols for patients with pre-existing bladder dysfunction, and how did these affect the incidence of post-transplant complications?

2.      What does the regular post-transplant follow-up involve, particularly in monitoring for urological complications?

3.      How were the risk factors (CAKUT, BK virus, etc.) assessed in your patient population, and did they have a direct correlation with the incidence of complications?

4.      The reported incidence of 15.7% for urological complications is consistent with existing literature, but further comparison to larger databases could strengthen the findings.

5.      Consider providing more detail on the duration of follow-up for patients, especially for those who developed late-onset complications, to better understand long-term outcomes.

6.      It would be helpful to include more detailed statistics on the outcomes of different intervention types (endoscopic vs. surgical) beyond just success rates, such as the need for repeat procedures.

7.      Highlighting how the multidisciplinary team's role contributed to better outcomes could enhance the discussion on collaborative care.

8.      Suggesting areas for future research, especially in evaluating long-term graft function in relation to urological complications, would be beneficial.

Author Response

Dear Reviewer,

here we send our responses to your comments and suggestions. 

Comment 1:  What were the specific management protocols for patients with pre-existing bladder dysfunction, and how did these affect the incidence of post-transplant complications? 

Response 1: In our study, patients with pre-existing bladder dysfunction underwent urodynamic evaluation before transplantation to assess bladder capacity, compliance, and voiding patterns. Management strategies included: bladder rehabilitation before transplantation (timed voiding, pharmacotherapy (such as anticholinergics), and intermittent catheterization to improve bladder storage and emptying. None of these patients underwent bladder augmentation or urinary diversion before transplantation (paragraph 2.3 Thorough pre-transplantation urological assessment, Materials and Methods section, page 3, line 137-153). More in detail, of these patients 5 developed ureteral stricture and 3 RVU; these complications were managed with urinary diversion in 3 out 8 patients (37.5%), bladder augmentation in one case (12.5%), ureteral stent positioning in 2 patients (25%) and percutaneous drainage in another patient (12.5%). At mean follow-up of 65 months, we did not observe any graft loss. One patient died due to Sars-nCOV2 complicated infection (see Results section). Unfortunately, we did not record the therapeutic management of our control patients, therefore we hereby report only the specific management of the group of patients who developed urological complications. 

Comment 2: What does the regular post-transplant follow-up involve, particularly in monitoring for urological complications? 

Response 2: We added a "Diagnostic Work-up" paragraph in the Materials and Methods section, describing our follow-up protocol (page 4, line 168-203). 

Comment 3: How were the risk factors (CAKUT, BK virus, etc.) assessed in your patient population, and did they have a direct correlation with the incidence of complications? 

Response 3: Thank you for your question. We included in the text a description of the protocol used in our Center for the assessment of risk factors, along with the relevant references. Additionally, we have conducted and presented an analysis of the risk factors, including CAKUT and BK virus, in both the study and the control group. This analysis evaluates the potential correlation between these factors and the incidence of complications (see Results section, paragraph 3.3, page 11-12, line 400-418).

Comment 4: The reported incidence of 15.7% for urological complications is consistent with existing literature, but further comparison to larger databases could strengthen the findings. 

Response 4: We improved the discussion regarding the varying incidence rates reported in the literature, incorporating studies based on large databases such as the North American Pediatric Renal Trials and Collaborative Studies (NAPRTCS), and explaining the possible reasons for discrepancies in findings (see Discussion, page 13, line 456-466).

Comment 5: Consider providing more detail on the duration of follow-up for patients, especially for those who developed late-onset complications, to better understand long-term outcomes. 

Response 5: Thank you for the suggestion. We reported our mean follow-up time in the first part of the Results section (in red, page 5, line 233-234). We also added a small paragraph detailing our follow-up protocol (see 2.5 Diagnostic Work-up paragraph in the Materials and Methods section, page 4, line 168-203).  

Comment 6: It would be helpful to include more detailed statistics on the outcomes of different intervention types (endoscopic vs. surgical) beyond just success rates, such as the need for repeat procedures. 

Response 6: Thank you for your suggestion. We added data in the Results, detailing the need for repeat procedures or the change of type of management (see Results section, paragraph 3.2, page 7-8, line 284-294 and 300-303). However, due to small sample size, no additional statistical analysis could be performed.  

Comment 7: Highlighting how the multidisciplinary team's role contributed to better outcomes could enhance the discussion on collaborative care. 

Response 7: According to your suggestion, we added a paragraph to the Material and Methods section of the manuscript detailing our pre-transplant urological assessment. This addition provides a comprehensive overview of the evaluation process that candidates to kidney transplantation undergo.We believe that this enhancement will clarify our methodology and reinforce the rigor of our assessment protocol (added text in red, paragraph 2.3 Thorough pre-transplantation urological assessment, Materials and Methods section, page 3, line 137-153). 

Comment 8: Suggesting areas for future research, especially in evaluating long-term graft function in relation to urological complications, would be beneficial. 

Response 8: Thank you for your suggestion. We added possible future perspectives at the end of the Discussion section (Discussion, page 16, line 625-628).  

Reviewer 3 Report

Comments and Suggestions for Authors

Dear authors you will find attached my comments.

Author Response

Dear Reviewer,

here we send our responses to your comments and suggestions. 

Comment 1:  Abstract: line 12 preferred  treatment, please rephrase. Transplantation is the treatment of choice. It is important to emphasize this. Line 14: 1-27% please explain the range.

Response 1: Thank you for the suggestions. We revised the sentence on line 12 to clarify the concept, as requested (in red, page 1, line 12). Regarding line 14, the range is derived from multiple studies, which are cited in the Discussion and reported in the References (reference 12-19). A comparison between the incidence of urological complications observed in our cohort and those reported in other studies was also added in the discussion section (Discussion,  page 13, line 456-466). 

Round 2

Reviewer 1 Report

Comments and Suggestions for Authors

Thank you for the answers and clarifications.

Comments on the Quality of English Language

The English could be improved to more clearly express the research.